# Screening Papaveraceae as Novel Antibiofilm Natural-Based Agents

**DOI:** 10.3390/molecules26164778

**Published:** 2021-08-06

**Authors:** Sylwia Zielińska, Magdalena Dziągwa-Becker, Adam Junka, Ewelina Piątczak, Anna Jezierska-Domaradzka, Malwina Brożyna, Justyna Paleczny, Aleksandra Sobiecka, Wojciech Słupski, Eleonora Mess, Mariusz Kucharski, Serhat Sezai Çiçek, Christian Zidorn, Adam Matkowski

**Affiliations:** 1Department of Pharmaceutical Biotechnology, Wroclaw Medical University, Borowska 211, 50-556 Wrocław, Poland; pharmaceutical.biology@wp.eu; 2Department of Weed Science and Tillage Systems, Institute of Soil Science and Plant Cultivation State Research Institute, Orzechowa 61, 50-540 Wrocław, Poland; m.dziagwa@iung.wroclaw.pl (M.D.-B.); m.kucharski@iung.wroclaw.pl (M.K.); 3Department of Pharmaceutical Microbiology and Parasitology, Wroclaw Medical University, Borowska 211a, 50-556 Wrocław, Poland; feliks.junka@gmail.com (A.J.); malwinabrozyna@gmail.com (M.B.); justyna.paleczny@student.umed.wroc.pl (J.P.); 4Department of Biology and Pharmaceutical Botany, Medical University of Łódź, Muszyńskiego 1, 90-151 Łódź, Poland; ewelina.piatczak@umed.lodz.pl; 5Department of Pharmaceutical Biology and Botany, Wroclaw Medical University, Borowska 211, 50-556 Wrocław, Poland; anna.jezierska-domaradzka@umed.wroc.pl (A.J.-D.); aleksandra.sobiecka@umed.wroc.pl (A.S.); 6Laboratory of Experimental Cultivation, Botanical Garden of Medicinal Plants, Wroclaw Medical University, Al. Jana Kochanowskiego 14, 50-556 Wrocław, Poland; 7Department of Pharmacology, Wroclaw Medical University, Jana Mikulicza-Radeckiego 2, 50-345 Wrocław, Poland; wojciech.slupski@umed.wroc.pl; 8Department of Oncology and Palliative Care, Wroclaw Medical University, K. Bartla 5, 51-618 Wrocław, Poland; eleonora.mess@umed.wroc.pl; 9Pharmazeutisches Institut, Abteilung Pharmazeutische Biologie, Christian-Albrechts-Universität zu Kiel, Gutenbergstraße 76, 24118 Kiel, Germany; scicek@pharmazie.uni-kiel.de (S.S.Ç.); czidorn@pharmazie.uni-kiel.de (C.Z.)

**Keywords:** *Chelidonium majus*, *Corydalis cava*, *Corydalis cheilanthifolia*, *Corydalis pumila*, *Fumaria vaillantii*, chelerythrine, coptisine, berberine, protopine, chlorogenic acid, quercetin

## Abstract

The antimicrobial properties of herbs from Papaveraceae have been used in medicine for centuries. Nevertheless, mutual relationships between the individual bioactive substances contained in these plants remain poorly elucidated. In this work, phytochemical composition of extracts from the aerial and underground parts of five Papaveraceae species (*Chelidonium majus* L., *Corydalis cava* (L.) Schweigg. and Körte, *C. cheilanthifolia* Hemsl., *C. pumila* (Host) Rchb., and *Fumaria vaillantii* Loisel.) were examined using LC-ESI-MS/MS with a triple quadrupole analyzer. Large differences in the quality and quantity of all analyzed compounds were observed between species of different genera and also within one genus. Two groups of metabolites predominated in the phytochemical profiles. These were isoquinoline alkaloids and, in smaller amounts, non-phenolic carboxylic acids and phenolic compounds. In aerial and underground parts, 22 and 20 compounds were detected, respectively. These included: seven isoquinoline alkaloids: protopine, allocryptopine, coptisine, berberine, chelidonine, sanguinarine, and chelerythrine; five of their derivatives as well as non-alkaloids: malic acid, trans-aconitic acid, quinic acid, salicylic acid, trans-caffeic acid, *p*-coumaric acid, chlorogenic acid, quercetin, and kaempferol; and vanillin. The aerial parts were much richer in phenolic compounds regardless of the plant species. Characterized extracts were studied for their antimicrobial potential against planktonic and biofilm-producing cells of *S. aureus*, *P. aeruginosa*, and *C. albicans*. The impact of the extracts on cellular metabolic activity and biofilm biomass production was evaluated. Moreover, the antimicrobial activity of the extracts introduced to the polymeric carrier made of bacterial cellulose was assessed. Extracts of *C. cheilanthifolia* were found to be the most effective against all tested human pathogens. Multiple regression tests indicated a high antimicrobial impact of quercetin in extracts of aerial parts against planktonic cells of *S. aureus*, *P. aeruginosa*, and *C. albicans*, and no direct correlation between the composition of other bioactive substances and the results of antimicrobial activity were found. Conclusively, further investigations are required to identify the relations between recognized and unrecognized compounds within extracts and their biological properties.

## 1. Introduction

The antimicrobial activity of Papaveraceae (poppy family) plants has been linked to the presence of several classes of alkaloids such as phthalideisoquinolines or tetrahydroprotoberberines and various polyphenolic compounds [1,2]. So far, most of the biological activities of these plants referred to the type of natural compounds or their concentrations [1,3]. Not much is known on the quiddity of individual compounds’ contribution in plant extracts that result in its biological potential. Can we discuss the strength of the biological activity in relation to the proportion between different bioactive metabolites or their classes? Papaveraceae are rich in alkaloids among which isoquinoline-derived structures constitute a large group. The family of Papaveraceae encompasses 44 genera and about 825 species of herbaceous plants distributed in the temperate zone of the northern hemisphere, in southern Africa, and in the western part of South America [4]. The APG IV system divides this family into three subfamilies and five tribes [4]. Many species of Papaveraceae have been used medicinally since ancient times. As medicinal raw materials, *Chelidonium majus*, *Fumaria* species, and *Corydalis* species were described by Dioscurides and Pliny the Elder in the first century AD, and under the names *Chelidonion*, *Capnos*, and *Aristolochia rotunda*, these species were known to physicians until the end of the Renaissance [2,5,6]. Some of them are still used as traditional herbal medicines. Each of the species represents its own phytochemical profile consisting of a varying number of alkaloids, different kinds of phenylpropanoids, and minor compounds such as terpenoids, saponins, and others. Regarding the alkaloids, it is the most diverse group of the phytochemical constituents, among which many derivatives with slightly different structures are observed [1,2]. The abundance of molecules of known or alleged antimicrobial potential within medicinal plants from Papaveraceae is of paramount importance in the present times, referred to as the “post-antibiotic era”, in which an increasing percentage of bacterial strains have become resistant to two or more classes of antibiotics [7]. Apart from resistance mechanisms, another adaptative microbial system is correlated with an increased risk of antibiotic therapy failure. The majority of pathogenic bacteria form an adhered structure referred to as the biofilm. This diversified community of microbial cells displays a high tolerance to antibiotics due to the presence of a protective extracellular matrix, diversified metabolism, and coordinated response systems [8]. The biofilm formed by such opportunistic pathogens as *Staphylococcus aureus*, *Pseudomonas aeruginosa*, and *Candida albicans* is known to have a highly detrimental impact on human health and it causes an increased tolerance to antibiotics in comparison to their “planktonic” counterparts (non-adhered, “free-swimming” cells) [9]. Bearing the above in mind, searching for novel antimicrobial and antibiofilm compounds is of great importance in order to increase the therapeutic success rate and decrease the economic burden related to ongoing, biofilm-based infections.

Therefore, in this work, we aimed to explore the similarities and differences in the phytochemical profile and the antimicrobial/antibiofilm potential of isoquinoline alkaloid-containing plants. We selected five species for the analysis, assigned to three different tribes within one plant family.

Plants were chosen for the phytochemical comparative analysis on the basis of differences in their morphology and anatomy. Two of the *Corydalis* species, *C. cava* (L.), Schweigg. and Körte, and *C. pumila* (Host), Rchb., develop bulbs as underground parts. One species, *C. cheilanthifolia*, Hemsl. known as the fern-leaved corydalis, develops tap roots, similarly to the representatives of two other genera, *Chelidonium majus* L. and *Fumaria vaillantii* Loisel. The detailed phytochemical composition of the aerial and underground parts was explored using LCMS equipped with a triple quadrupole detector. Next, assays measuring the impact of extracts against planktonic and biofilm-forming *S. aureus*, *P. aeruginosa*, and *C. albicans* were performed. Finally, the extracts were chemisorbed within a polymeric carrier, referred to as the bacterial cellulose (BC), to show, as the proof-of-concept, the potential applicability of extracts as antimicrobial measures against localized infections.

## 2. Results

### 2.1. Phytochemical Analysis

Large differences in the quality and quantity of all analyzed compounds were observed between species of different genera and also within one genus. Isoquinoline alkaloids predominated in the phytochemical profiles and several phenolic compounds were also present in smaller quantities.

#### 2.1.1. Qualitative Analysis

A total of 22 compounds in aerial parts and 20 compounds in underground parts of the five Papaveraceae species were detected. Twelve of them were identified in positive and another ten in the negative electrospray ionization mode.

In extracts of the aerial and underground parts of *C. majus, C. cava*, *C. cheilanthifolia*, *C. pumila*, and *F.* vaillantii, the following compounds were detected: seven isoquinoline alkaloids: protopine, allocryptopine, coptisine, berberine, chelidonine, sanguinarine, and chelerythrine; five of their derivatives: protopine derivative, coptisine derivative, tetrahydrocoptisine, tetrahydroberberine, and chelidonine derivative; three non-aromatic carboxylic acids: malic acid (dicarboxylic acid), trans-aconitic acid (polycarboxylic acid), and quinic acid (cyclohexanecarboxylic acid); one hydroxybenzoic acid: salicylic acid; two hydroxycinnamic acids: trans-caffeic acid and *p*-coumaric acid; one caffeic and the quinic acid ester: chlorogenic acid; two flavonols: quercetin and kaempferol; and one phenolic aldehyde: vanillin.

The assignment of the protopine derivative was based on a characteristic loss of the *m*/*z* 320.2 fragment attributed to protonated protopine (Table 1 and Table 2). Allocryptopine exhibited the parent ion at *m*/*z* 369.6 and the product ions at *m*/*z* 352, 187.9, 290. The chelidonine derivative showed the precursor ion at *m*/*z* 370 and the product ions at *m*/*z* 356 and 339. The most abundant precursor ion at *m*/*z* 340 was assigned for tetrahydroberberine and at *m*/*z* 336.4 for berberine. Two of the coptisine derivatives showed the parent ion at *m*/*z* 324, while for coptisine at *m*/*z* 320.1. Sanguinarine and chelerythrine assignment was based on a characteristic loss of a fragments of *m*/*z* 332.1 and 348.1, respectively.

The presence of three carboxylic acids was detected on the basis of parent ions corresponding to malic acid (*m*/*z* 133.1), trans-aconitic acid (*m*/*z* 172.9), and quinic acid (*m*/*z* 191). Hydroxybenzoic acid showed parent ions at *m*/*z* 137.3 corresponding to salicylic acid. Trans-caffeic acid and *p*-coumaric acid identification was based on parent ions at *m*/*z* 179.2 and 163, respectively. Chlorogenic acid exhibited the parent ion at *m*/*z* 353 and a characteristic loss of fragments corresponding to caffeic and quinic acids. The identification of two flavonoids was based on the presence of the parent ion at *m*/*z* 301.1 corresponding to quercetin and at *m*/*z* 285 to kaempferol. One phenolic aldehyde showed the parent ion at *m*/*z* 151.2 and the product ions at *m*/*z* 136, 91.8, 108 corresponding to vanillin (Table 1 and Table 2).

#### 2.1.2. Phytochemical Quantitative Analysis

In aerial parts of all five species allocryptopine was found in quantities ranging between 5.96 and 265.06 µg/g of dry weight (D.W.), whereas in underground ones between 9.26 and 699.72 µg/g of D.W. (Table 1 and Table 2).

Protopine was found in aerial parts of all plants in the range between 136.52 and 1083.21 µg/g of D.W. In underground parts, this compound was present in all studied species in the wide range of 131.79–6529.64 µg/g of D.W. A protopine derivative was present in all the studied species in both plant organs, except for the aerial parts of *C. pumila*. The amounts of this compound remained unknown because the identification was based only on mass spectra without an authentic reference substance (Table 1 and Table 2).

Coptisine was detected in relatively large amounts compared to other alkaloids and polyphenolic compounds. Its highest content was found in the aerial parts of *C. cheilanthifolia*, followed by *C. majus*, *C. cava*, *F. vaillantii*, and *C. pumila* (5794.90, 5455.79, 1820.82 µg/g, 76.73 µg/g, and 2.75 µg/g, respectively). Relatively large amounts of this compound were also present in the underground parts of *C. cheilanthifolia* and *C. majus* (3605.56 and 2744.21 µg/g, respectively). In turn, in both the aerial and underground parts of *C. cava*, *C. pumila*, and *F. vaillantii*, coptisine was found in much lower amounts (Table 1 and Table 2). Tetrahydrocoptisine and an unidentified coptisine derivative were detected in all samples but their absolute quantities remained unknown due to their identification being based only on mass spectra with no reference substances (Table 1 and Table 2).

Berberine was abundant in aerial parts of *C. cheilanthifolia* (2696.04 µg/g of D.W.) and in the underground parts of *C. pumila* and *C. majus* (739.03 and 345.45 µg/g, respectively). In the other species, it occurred only in much smaller amounts (Table 1 and Table 2). In turn, tetrahydroberberine was found in all samples but not quantified due to the lack of the reference substance (Table 1 and Table 2).

Chelidonine was found in *C. majus* in quite large amounts of 130.77 and 1936.11 µg/g in aerial and underground parts, respectively. Smaller quantities of this compound were found in underground parts of *Fumaria vaillantii* (48.58 µg/g of D.W.), *C. cava* (13.69 µg/g of D.W.), and *C. cheilanthifolia* (6.74 µg/g of D.W.), whereas in *C. pumila* it was not detected. In the aerial parts of *C. cava*, *C. pumila*, and *F. vaillantii*, it was present in the range of 0.24–8.03 µg/g of D.W. and not detected in *C. cheilanthifolia*. The Chelidonine derivative was present only in the aerial parts of *C. majus* and *C. cheilanthifilia*, and in the underground parts of *C. majus*, *C. cava*, and *C. pumila* (Table 1 and Table 2). No quantitative analysis was performed for this compound.

The largest amount of chelerythrine was found in the underground parts of *C. majus* (742.36 µg/g of D.W.). In the remaining samples it was detected in smaller amounts (1.25–48.28 µg/g of D.W.) (Table 1 and Table 2).

Sanguinarine was found in large amounts in underground parts of *C. majus* and *C. cheilanthifolia* (915.56 and 493.20, respectively), and in the rest of plant samples in the range of 2.53–57.22 µg/g of D.W. depending on the plant species (Table 1 and Table 2).

Generally, much lower amounts of phenolic compounds were found compared to the amounts of alkaloids and they were mainly present in the aerial parts. Salicylic acid was found in a detectable amount of 3.05 µg/g of D.W. only in aerial parts of *C. majus*. (Table 1). Trans-caffeic acid, chlorogenic acid, and *p*-coumaric acid were present in aerial parts of almost all the plants in the amounts of 16.05–130.37, 82.37–750.97, and 1.05–78.89 µg/g of D.W., respectively (Table 1). Additionally, in the underground parts of *C. cava*, a considerably large quantity of chlorogenic acid was found (1198.03 µg/g of D.W.) (Table 2).

Vanillin was found only in aerial parts of *C. majus* (15.54 µg/g of D.W.) and in underground parts of *C. pumila*, *C. cheilanthifolia*, and *F. vaillantii* in amounts of 12.4, 28.93, and 18.71 µg/g of D.W., respectively (Table 2).

Quercetin was detected in aerial parts of all the examined species in amounts between 1.23 and 262.36 µg/g of D.W., with the highest content in *C. cheilanthifolia*. In the underground parts, it was found only in *C. cheilanthifolia* and *F. vaillantii* (205.54 and 305.65 µg/g of D.W., respectively) (Table 2). In aerial parts of the former species, another flavonoid, kaempferol, was detected but its content was at the limit of quantitation (Table 1).

Other polyphenolic compounds were mainly present in the aerial parts of the studied plants in quantities that did not allow for their quantification.

### 2.2. The Antimicrobial Assays

The results of antimicrobial assays against planktonic cells showed that the extract of the *C. cheilathifolia* herb acted not only against all tested pathogens (*C. albicans*, *P. aeruginosa*, and *S. aureus*), but was also the most effective in the aspect of the concentration applied (MIC at 50%, 25%, and 6.25%, respectively) (Figure 1A). The minimum inhibitory concentrations of other extracts were not lower than 50% or the MIC was not reached in a concentration of least 50% against all microbial strains, with one exception of *C. cava* bulbs (MIC at 25% against *S. aureus*) (Figure 1A). Conversely, all extracts that did not reach the MIC threshold still displayed noticeable reduction rates against analyzed microbes (Figure 1B–D). In the cases of *S. aureus*, *P. aeruginosa*, and *C. albicans*, reduction rates for these extracts oscillated around 30‒70%, 40‒80%, and 70‒80%, respectively.

The investigations on the antibiofilm activity showed a higher impact of the analyzed extracts on the number of *S. aureus* and *C. albicans* biofilm-forming cells (and their metabolic activity) than on the biofilm biomass (which comprises of cells and the biofilm extracellular matrix). Although the highest-to-achieve concentration of extracts (50% (*v*/*v*)) did not lead to the complete (100%) killing of *S. aureus* and *C. albicans* cells (thus it may not be considered the minimal biofilm eradication concentration), the reduction rate in the case of *C. albicans* was high (above > 92%). It is also noteworthy that the opposite trend was observed in the case of *P. aeruginosa*; the application of extracts repeatedly increased the metabolic cellular activity of the biofilm of this pathogen. The high diversification was observed with regard to the part of the plant applied. In the case of *S. aureus*, the underground parts’ extracts of only two species (*C. majus* and *F. vaillantii*) were more effective than those of the aerial parts (Table 3). In turn, the herb extracts of *C. cheilathifolia*, *C. pumila*, and *C. cava*, used in the concentration of 50%, more effectively eradicated *S. aureus* cells than the underground parts’ extracts (Table 3). Results of the multiple regression test (Table 4) indicated that, with regard to planktonic cells, quercetin had the highest impact on the results obtained against all microorganisms analyzed. No similar tendency (high impact of one compound against all microorganisms) was observed in other experimental settings (underground parts vs. planktonic cells or biofilms). This suggests an influence of the compounds’ mixture on antimicrobial results. Nevertheless, in the case of *P. aeruginosa*, the allocryptopine extracted from underground parts should be distinguished, as its presence correlated with antimicrobial results against this pathogen (Table 4). The antimicrobial activity of extracts released from bacterial cellulose carrier, used during examination, indicated *C. cava* and *C. cheilathifolia* to be the most effective against *S. aureus* and *C. albicans* (Table 5).

## 3. Discussion

The comparative analyses of the chemical composition of the selected plant species and the results of their antimicrobial activity indicated that the biological potential of these plants does not depend solely on the content of any individual constituent. The studied species represent two subfamilies within the Papaveraceae, i.e., Fumarioideae Eaton with the genera *Fumaria* L. and *Corydalis* D.C., and Papaveroideae (*Chelidonium majus*). Fumarioideae, unlike Papaveroideae, do not produce latex, although laticifer tubes are developed in collateral vascular bundles, the cortex, and the sclerenchymatous sheath of *Fumaria densiflora* stem [10]. About 400 species belong to the *Corydalis* genus. These are plants widespread in Europe, Asia, North America, and northern, eastern, and southern Africa. Most species are native to China and the Himalayas [4]. They are usually perennials that develop racemose inflorescences with monosymmetric flowers and underground bulbs. However, some species develop a tap root.

In this study, the extracts of three *Corydalis* species were the most effective against *S. aureus* and *Candida* sp., among which both aerial parts and root extracts of *C. cheilanthifolia* had the strongest antimicrobial effect. Simultaneously, these extracts were rich in isoquinoline alkaloids such as protoberberine and protopine derivatives (Table 1 and Table 2). The aerial parts also contained relatively large amounts of chlorogenic acid, quercetin, and kaempferol, and smaller amounts of trans-caffeic acid and *p*-coumaric acid (Table 1 and Table 2). Previous studies on this species revealed the presence of the same alkaloids [11]. In addition, other alkaloids were detected such as ophiocarpine, (−)-canadine, (−)133-hydroxystylopine, (−)-canadine methohydroxide, and stylopine, as well as trans-ferulic acid. However, the mentioned study was only qualitative and used only the aerial parts. The results of our analysis additionally indicated the presence of phenanthridine derivatives such as sanguinarine and chelerythrine, as well as quercetin, chlorogenic acid, and vanillin in the roots (Table 1 and Table 2). *C. cheilanthifolia* is a short-lived perennial native to the western provinces of China, growing on shady slopes at an altitude of 800–1700 m above sea level. Unlike many other *Corydalis* species, it does not produce bulbs, only a tap root. 

It is interesting that the second most effective were the two other *Corydalis* species, *C. cava* and *C. pumila* (Table 1 and Table 2). *C. cava* is an early spring geophyte associated with European oak-hornbeam forests. The bulb of this species is large with roots growing all over its surface and hollow in older individuals. The *C. cava* phytochemical profile was rather poor in terms of the complexity and quantity of alkaloids and phenolic compounds, with an exception of chlorogenic acid (Table 1 and Table 2). A similar but only qualitative phytochemical profile of the aerial parts of *C. cava* was observed by Adsersen et al. [12] based on thin-layer chromatograms. Allocryptopine was also found in *C. cava* by Slavik and Slavikova [13], while Och et al. [14] did not detect this compound at all. Additionally, a secoberbine alkaloid, (−)-canadaline, was isolated and identified in the bulbs of *C. cava* by Meyer and Imming [15]. The molecular weight and molecular formula are the same as for allocryptopine. This may cause discrepancy between different studies. We did not detect (−)-canadaline in the examined plant material, while allocryptopine was identified using the reference substance. Coptisine was previously identified in *C. cava* bulbs by Och et al. [14] in a moderate relative content of 15–30% based on the peak area at 270 nm with no reference standard. It was also isolated in small amounts from bulbs by Slavik and Slavikova [13] and Gasic et al. [16]. We have detected this compound only in aerial parts, while Preininger et al. [17] noted it as a minor compound. In 2012, a new alkaloid named corylucinine was also isolated from *C. cava* bulbs [18]. It may be involved in bioactivity but we have not detected it in our plant material.

In turn, in the phytochemical profile of *C. pumila*, aerial parts were dominated by protopine and in the underground parts by allocryptopine, berberine, and protopine (Table 1 and Table 2). Both plant parts were poor in phenolic compounds. Only small amounts of caffeic acid, *p*-coumaric acid, and quercetin were found (Table 1 and Table 2). Unfortunately, no other studies could be found to compare with our results of *C. pumila*, a sub-Atlantic species found in Europe from southern Scandinavia and through Central Europe to the Balkans and Corsica. In Poland, it is a vulnerable (VU) species growing only in a few localities in the western part of the country. It grows in mixed forests, most often in moist oak-hornbeam forests. The bulb is small, solid inside, and renews every year. In the spring, the old bulb cambium sets a new bulb and the old one is reduced to thin structures surrounding the new bulb [19].

Two further species (*C. majus* and *F. vaullantii*) showed moderate and low antimicrobial activity. The extracts from the underground parts showed stronger antibacterial and antifungal activity against planktonic and biofilm cells. Roots of *F. vaillantii* were characterized by a relatively rich phytochemical profile (Table 2). They contained alkaloids from three classes, i.e., protopine, protoberberine, and phenanthridine derivatives, and their content was several times higher than in the aerial parts (Table 1). Previous studies of this species have shown the presence of several classes of isoquinoline alkaloids such as benzylisoquinoline, spirobenzylisoquinoline, phthalideisoquinoline, morphinandiene, protoberberine, protopine, and benzophenanthridine derivatives [15,20,21,22,23,24,25,26,27,28], as well as quercetin [29]. In addition, several other classes of phytochemicals were detected using obsolete unspecific methods [30]. Among different compounds, fumaric acid found in *F. vaillantii* was indicated as potentially responsible for the antimicrobial activity [30] but no other compounds were examined in this reference. *F. vaillantii* grows in Europe, Turkey, the Middle East, the Caucasus, Iran, Central Asia, northwest Africa, and Libya [31]. It is an archeophyte originating in the Irano-Turan area. In Poland, it is a rare species and is classified as vulnerable (VU). Due to its morphologic resemblance to the pharmacopeia-listed species *Fumaria officinalis*, there is a risk of misidentification of the herbal material when collected from wild habitats. Therefore, the data on their phytochemical profiles are important for pharmaceutical quality.

*C. majus* roots were found to be the richest source of isoquinoline alkaloids in terms of their quality (Table 2). Seven alkaloids of various classes and their derivatives have been found in both aerial and underground parts. The former contained significantly lower amounts of phenanthridine derivatives. The content of sanguinarine and chelerythrine in the aerial parts was several dozen times lower than in the roots (Table 1 and Table 2). In turn, the coptisine content was twice as much as in the roots (over 5 mg/g, Table 1). Most of the previously published papers on *C. majus* concern almost exclusively the presence of isoquinoline alkaloids. However, the raw material harvest period is often not defined, thus the alkaloid contents are presented in considerably varying amounts. This issue has been discussed in details in a recent review by Zielińska et al. [2]. Our current research showed that both the *C. majus* herb and its roots harvested in April are rich in coptisine. In the previous one [3], such large amounts of coptisine were noted only for aerial parts, especially fruits, but the plant material was harvested in autumn (September). A large proportion of this compound was also detected in shoot cultures of *C. majus*, while root cultures were rich in sanguinarine [32]. Larger proportions of chelidonine, chelerythrine, and sanguinarine in roots were also observed by Tome and Colombo [33]. Considering polyphenolic compounds in *C. majus*, only a few reports in the literature exist. In our study, the *C. majus* herb contained several polyphenolic compounds, among which hydroxycinnamic acids such as trans-caffeic acid, *p*-coumaric acid, and chlorogenic acid were the most abundant (Table 2, Figure 2). Caffeic, *p*-coumaric, *p*-hydroxybenzoic, caffeoyl-threonic acid, caffeoyl-glyceric acid, and caffeoyl-malic acid were previously identified in *C. majus* [34,35,36]. Additionally, in the study by Grosso et al. [34] two flavonoids, i.e., quercetin and kaempferol and their derivatives, were detected. Genus *Chelidonium* L. belongs to the tribe Chelidonieae Dumortier of the subfamily Papaveroideae Eaton and is comprised of only two species: *C. majus* and *C. asiaticum* (Hara) Krahulc. [4,37]. *C. majus* is a short-lived perennial native to temperate areas of Europe and Asia (Erhardt et al., 2002). The natural habitat of *C. majus* are deciduous forests, brushwood, and ruderal sites with a high content of organic matter and nitrogen in the soil. Both herbs, as well as the roots of celandine, produce orange latex, visible after injury to the plant. As a pharmacopeial species growing wild and cultivated in many regions, knowledge of the phytochemical diversity and correlating it to potential bioactivity is of high importance.

When considering the content of individual compounds, allocryptopine was found in all five examined species, although its amounts were higher in underground parts (Table 1 and Table 2). Protopine and coptisine were the metabolites with the largest quantitative differences between the examined plant species. Unfortunately, due to the lack of reference substances, it was impossible to identify these compounds more accurately and quantify them.

The analysis of the polyphenolic compounds generally showed that they were more characteristic for the aerial parts of the studied plants (Table 1, Figure 2). However, it is difficult to clearly indicate how the phytochemical profile of the studied plants translates into their antimicrobial activity. The assays against planktonic *C. albicans* and *S. aureus* revealed a stronger effect of the extracts from the underground parts of all the studied species and in *C. cheilathifolia*, *C. pumila*, and *F. vaillantii* against *P. aeruginosa*.

In turn, the stronger effect of the extracts from the underground parts against planktonic *P. aeruginosa* cells was visible in the case of three out of five plants tested (*C. cheilathifolia, C. pumila*, and *F. vaillantii*). The underground parts of *C. majus* and *F. vaillantii* and aerial parts from all *Corydalis* species were more effective against the *S. aureus* biofilm (Figure 1, Table 3).

If one of the identified compounds was identified as having the highest impact alone on the antimicrobial result, it would be quercetin but only with regard to planktonic *P. aeruginosa* cells. This coincides with the recent work of Kho et al. [38]. However, contrary to aforementioned work, our results do not indicate the antibiofilm potential of quercetin against *P. aeruginosa*. This apparent discrepancy may be explained by the fact that quercetin acts in a strain-dependent manner and its concentration may be too low to eradicate the biofilm of this particular reference pseudomonal strain we used in our study. However, various phenolic compounds including flavonols and phenolic-rich extracts are often reported for their general antimicrobial properties [39,40,41,42]. It is noteworthy that the results concerning the biofilm of *P. aeruginosa* indicate an increase of the metabolic activity of cells (Table 3) after exposure to extracts. This phenomenon requires further experimental studies; at the moment, a few hypotheses (requiring empirical confirmation) can be proposed. The first of them is related to the stimulating impact of so-called sub-optimal (referred to as sub-MIC) antimicrobials’ concentrations on the metabolic activity of pseudomonal biofilm shown, among others, by the team of Hemati et al. [43]. The second one is related to the reported ability of members of the *Pseudomonadaceae* family (referred to as the *P. entomophila*) to secrete isoquinoline alkaloids similar in chemical structure to berberine and sanguinarine from Papaveracae [44]. Considering the high genetic similarity of *Pseudomonadaceae* members [45], it may be assumed that together with the ability to produce isoquinoline alkaloids, aforementioned bacteria including the *P. aeruginosa* species developed also specific mechanisms to neutralize alkaloids’ biocidal impact against its own cellular structures. Their activation and subsequent energy consumption may translate into observed high metabolic activity. The third hypothesis assumes that specific molecules or their mixtures from the analyzed samples are responsible for the up-regulation of one of the energy-producing metabolic pathways within pseudomonal cells. If it is so, such specific molecules or molecules’ mixtures could be possibly applied in the biotechnological industry, utilizing pseudomonal cells for a number of applications including biosurfactant production [46]. 

In regard to the underground parts, the applied multiple regression model (Table 4) indicated that allocryptopine was a molecule of significant potential for the eradication of bacterial (*P. aeruginosa*, *S. aureus*) cells but not of *C. albicans*. Nevertheless, all other molecules than the already mentioned quercetin and allocryptopine, which had an impact on the observed reduction of the microbial count, were of known antimicrobial activity. These include chlorogenic acid, sanguinarine, berberine, coptisine, *p*-coumaric acid, chelerythrine, chelidonine, allocryptopine, sanguinarine, and vanillin (Table 4). These compounds act through a variety of mechanisms of which two groups should be distinguished, namely the destruction of the bacterial membrane or DNA replication inhibition groups [47,48,49,50]. The above results show that the observed antimicrobial effect is predominately mediated by the composition of active molecules within extracts and their mutual interactions that may occur during the process of interplay with microorganisms. Therefore, it seems to be crucial in the next line of investigation to analyze the activity of all these molecules alone and coupled with assessing which applied combination acts in an antagonistic and synergistic manner with regard to observed antimicrobial effect. 

The results presented in this work show the promising potential of Papaveracae plants to be applied as a source of novel antimicrobial and antibiofilm drugs, and simultaneously demonstrate that specific microbial species (*P. aeruginosa*) may exhibit contrary types of responses to extracts, i.e., to increase metabolism (Table 3). It may affect the therapeutic indications and can potentially cause adverse effects if used against unsusceptible bacterial strains. Therefore, further study is necessary to verify the often-seen claims of general or wide-ranging antibacterial properties of berberine and related alkaloids. There are also other limitations of this study that require further action before there is a direct translation of the obtained results into the clinical setting. 

The first of them is the number of microbial strains analyzed. Undoubtedly, the analysis of a few dozen of strains from each species would allow to scrutinize applied extracts with regard to the inter and intra-species tolerance to active compounds present in extracts and shed light on their exact potential applicability in fighting against the infection process. The second of them is related to results presented in Figure 2, Table 5. One should note that the majority of extracts released from the BC carrier were ineffective against lawn-seeded cells, in contrast to the results presented in Figure 1 and Table 3. It may indicate the interactions (bonding) between the applied carrier and bioactive molecules of extracts. The application of another carrier or functionalization of the BC carrier would be necessary if such bonding was confirmed in subsequent studies. The third disadvantage is of a strictly methodological character and concerns the methods of the pseudomonal biofilm culturing in a 96-well plate setting. As we and other teams reported earlier [51,52,53,54,55], the commonly acknowledged microtiter plate methods, although applicable for the testing of biofilms localized and firmly adhered to the bottom of the plate well (as in the case of *S. aureus* or *C. albicans*), are of low applicability for analyses of slime-forming (and filling the whole volume of the plate well) biofilms of the *Enterobacteriacae* or *Pseudomonas* family. The subsequent procedures of rinsing and washing that are necessary to perform the crystal violet assay often lead to the random removal of the pseudomonal biofilm and to bias the results of high standard deviations. Therefore, although we performed such a test, we deliberately did not include it in the manuscript. In turn, we presented data from the metabolic (TTC) assay for analyses of the pseudomonal biofilm because this technique allows to avoid the aforementioned disadvantages to some extent. As it was shown by Grela et al. [56], the tetrazolium-based test, if performed with precautions, allows not only to localize and quantify the biofilm structure but also to determine the presence of live bacteria within. Despite the above-mentioned limitations, the results of this study indicate the antimicrobial or microbiologically modulating potential of the high number of bioactive molecules isolated from Papaveracae plants, which are of high application for further studies aiming to discover the new non-antibiotic counter-measures useful for the eradication of nosocomial strains.

## 4. Materials and Methods

### 4.1. Plant Material

Five different plant species from the Papaveraceae family were collected from the Botanical Garden of Medicinal Plants in Wroclaw (BGMP), geographical location: 51°07′01.6” N 17°04′26.7” E 51.117121, 17.074088. These were *Chelidonium majus, Corydalis cava, Corydalis cheilanthifolia, Corydalis pumila*, and *Fumaria vaillantii*. Detailed edaphic data on the plants’ locations are as follow: *Corydalis cava*-soil with a thick layer of humus, well-drained, not moist, not sandy, with a small amount of debris; *Corydalis pumila*-soil with a thick layer of humus, well-drained, not moist, not very sandy; *Corydalis cheilanthifolia*-sandy soil, well-drained, with very low humus content; *Fumaria vaillantii*-anthropogenic soil with debris, with a thin humus layer on the surface, permeable; and *Chelidonium majus*-soil, very sandy, well-drained, slightly moist with a thin humus layer. Meteorology data obtained from the Polish Institute of Meteorology and Water Management, National Research Institute (IMGW Wrocław-Strachowice station) for April 2018 include: the average daily temperature of 14.0 °C; total rainfall of 31.4 mm; insolation of 284.3 h; absolute maximum daily air temperature (at a height of 2 m) of 27.2 °C; absolute minimum daily air temperature (at a height of 2 m) of −1 °C; absolute minimum daily air temperature at the ground (at a height of 5 cm) of −3.8 °C; and data indicating two days with frost occurring in the first week of April 2018.

Plants were harvested in spring during the flowering period between 10th and 27th of April 2018 (*C. majus* on 04/27; *C. cava*, *C. cheilanthifolia*, and *C. pumila* on 04/10; and *F. vaillantii* on 04/21). Voucher specimens of the species were deposited in the Herbarium of Botanical Garden of Medicinal Plants, Wroclaw Medical University, under the codes: L02/C.m.1–15/18, L01/C.c.1–9/18, L01/C.ch.1–11/18, L01/C.p.1–11/18, and L01/F.v.1–10/18.

Whole plants were dried at the temperature of 35 °C, separated into aerial and underground parts, and intended for extraction.

#### Plant Material Extraction

Dried aerial and underground parts (200 mg) of the plants were separated, grounded to powder using a mortar and pestle, and extracted with pure methanol and 0.1% formic acid using an ultrasonic bath (2 × 30 min) in a total volume of 5 mL. The extracts were prepared according to the modified procedure used in our previous studies [57,58].

### 4.2. Phytochemical Analysis

#### 4.2.1. Chemical and Reagents

LC-MS grade methanol, water, and an eluent-additive LC-MS ultra-ammonium formate (NH_4_HCO_2_) were purchased from Fluka Analytical (St.Louis, MO, USA).

Berberine (purity ≥ 95%), protopine (purity ≥ 95%), sanguinarine (purity ≥ 90%), chelerythrine (purity ≥ 90%), and chelidonine (purity ≥ 95%) were purchased from Extrasynthese (Genay, France), while coptisine (purity ≥ 98%), allocryptopine (purity ≥ 98%), tr-caffeic acid, *p*-coumaric acid, chlorogenic acid, vanillic acid, salicylic acid, vanillin, quinine sulphate, and quercetin were purchased from Sigma-Aldrich (St. Louis, MO, USA).

#### 4.2.2. Liquid Chromatography

For the identification and quantification of the compounds, we used liquid chromatography-electrospray ionization-tandem mass spectrometry (LC-ESI-MS/MS) with a triple quadrupole analyzer. The chromatographic separation was performed using a Shimadzu Prominence UFLC system (Shimadzu, Kyoto, Japan) equipped with a LC-30 ADXR binary solvent manager; DGU-20A3 degasser; CTO-10ASVP column oven; Kinetex C18 column, 2.6 μm particle size, 100 × 3.0 mm (Phenomenex, Torrance, CA, USA); SIL 20AXR autosampler; and CBM-20A system controller interfaced to a triple quadrupole analyzer. The separation was performed using reversed phase conditions. The mobile phase consisted of mixture A–B: 10 mM ammonium formate in water (A) and 0.1% formic acid in methanol (B). The methanol percentage was changed linearly as follows: 0 min, −10%; 10 min, −85%; 13.01 min, −85%; and 16 min, −10% at a flow rate of 0.40 mL min^−1^. The column temperature was maintained at 35 °C. The sample volume (10 mL) was injected in the UHPLC system.

#### 4.2.3. Mass Spectrometry

The tandem mass spectrometer LCMS-8030 (Shimadzu, Kyoto, Japan) with a triple quadrupole analyzer equipped with an electrospray ionization (ESI) source working in both positive and negative ionization modes was used for the analysis of alkaloids and polyphenolic compounds.

Nitrogen dried and nebulized gas obtained from pressurized air in a N2 LC-MS pump, working at a flow rate 15 L min-1 and 3 L min-1, respectively. The desolvation line temperature was 250 °C and the heat block temperature was 400 °C. The collision induced dissociation gas (CID) was argon 99.99% (Linde, Wrocław, Poland) at a pressure of 230 kPa. A dwell time of 10 ms was selected. LabSolution Ver. 5.6 (Shimadzu, Kyoto, Japan) software was used to process the quantitative data.

#### 4.2.4. Identification and Quantification

A full scan together with MS/MS spectra was obtained during the flow injection analysis (FIA) of each reference substance. The identification and quantification of analytes was performed using a multiple reaction monitoring (MRM) mode. The compounds’ identification was accomplished with the ion intensity ratio and retention time compared with the corresponding standards and the previously identified alkaloids and phenolic acids reported in the literature (for compounds 1, 5, 8, 9, and 10, shown in Table 1 and Table 2 from Grosso et al. [34]; for compounds 13 and 14 from Bylund et al. [59]; and for compound 15 from Erk et al. [60]). The content of the compounds was expressed in μg per g of plant material dry weight (D.W.).

The limit of the compounds’ detection (LOD) was calculated according to a signal-to-noise ratio (S/N) of 3 and the limit of the compounds’ quantitation (LOQ) was calculated according to the S/N ratio of 10. Linearity was evaluated from the square correlation coefficients (r2) of the regression curves performed for each reference substance (r2 ≥ 0.99 was achieved for all compounds).

### 4.3. Antimicrobial Assays

Prior to the performance of the antimicrobial tests, dried plant extracts were diluted with methanol and 0.1% formic acid (1:4, *v*/*v*).

#### 4.3.1. Assessment of the Minimal Inhibitory Concentration of the Analyzed Extracts

The antimicrobial activity of the extracts was determined using the MIC (minimum inhibitory concentration) assessment. Three reference strains from the American Tissue and Cell Culture Collection were analyzed: *Staphylococcus aureus* ATCC 6538, *Pseudomonas aeruginosa* ATCC 15442, and *Candida albicans* ATCC 10231. Initially, 100 μL of TSB (Trypticasein Soy Broth, Biomaxima, Lublin, Poland) medium was added to all the wells of a 96-well microtitration plate (Nest, China) except the first column. Subsequently, 200 μL of the extracts were added to the empty wells and geometrically diluted. Next, 24 h bacterial/fungal liquid cultures were adjusted to the 0.5 McFarland (MF) standard and diluted in TSB medium to 10^5^ colony-forming units (CFU)/mL on the basis of a previously performed calibration curve to mimic the concentration of microbes able to cause the adverse effects on human health. Then, 100 μL of the suspensions were poured into each well and the plate was incubated at 37 °C, shaken (400 rpm/min.) for 24 h. Control of the microorganisms’ growth (medium with cells) and antimicrobial activity of the methanol (Chempur, Poland) and 0.1% formic acid (Chempur, Poland) mixture, in a ratio of 1:4 (*v*/*v*), samples were also performed. Subsequently, 20 μL of the 1% solution of triphenyl tetrazolium chloride (TTC, 2,3,5-triphenyl-2H-tetrazolium chloride, Sigma Aldrich, Darmstadt, Germany) was added to the wells and incubated for 2 h at 37 °C. The MIC values were assessed as the first wells in which no red color was observed. Moreover, absorbance was measured before and after incubation at the wavelength *λ* = 580 nm using a spectrophotometer MultiScan Go (Thermo Fischer Scientific, Waltham, MA, USA). The absorbance measurements were performed in two separate experiments and backed-up with a visual assessment of the TTC reduction into red formazane. The results are presented as a precentage of cell reduction calculated by the formula:Cell reduction [%] = 100% − [Ab E/Ab PC × 100%](1)Abbreviations: Ab E—absorbance of samples measured after incubation cells with extracts and Ab PC—absorbance of samples measured after incubation cells with medium only.

Furthermore, 5 μL of the solution was taken from each well and cultured onto the Mueller–Hinton agar plates (Graso, Poland). The plates were incubated for 24 h at 37 °C and the MBC (minimum bactericidal concentration) values were evaluated as the first well in which no growth was observed.

#### 4.3.2. Assessment of the Antibiofilm Activity of the Analyzed Extracts

The antibiofilm activity of the 2-fold diluted extracts was assessed. Firstly, 100 μL of the bacterial/fungal suspensions prepared as described above were introduced to a 96-well microtitration plate (Nest, China) containing 100 μL of TSB (Trypticasein Soy Broth, Biomaxima, Lublin, Poland) medium in each well. After the plate incubation for 24 h at 37 °C, the medium was replaced with 200 μL of 50% concentration of the extracts, medium (positive control), or the solvent and incubated for 24 h at 37 °C. Then, the medium was removed again. Next, the number of metabolically active cells and the amount of formed biofilm were evaluated using Richard’s technique and the crystal violet biofilm assay, respectively. The first mentioned test was conducted as follows: 200 μL of 0.1% TTC (2,3,5-triphenyl-2H-tetrazolium chloride, Sigma Aldrich, Darmstadt, Germany) solution was added and the plate was incubated for 2 h at 37 °C. The medium was replaced with the same volume of methanol (Chempur, Poland) and the plate was shaken at RT for 30 min. 100 μL of the solution was transferred to a fresh plate and the absorbance was measured at the wavelength λ = 490 nm (spectrophotometer MultiScan Go (Thermo Fischer Scientific, Waltham, MA, USA).

To quantify the biofilm formation using crystal violet methodology, the plate was dried for 10 min at 37 °C. Then, 100 μL of 20% solution of crystal violet (CV, Aqua-med, Poland) was poured into the plate and incubated for 10 min/RT. After the dye was removed, the biofilm was washed twice with 100 μL of saline and the plate was dried for 10 min. Next, 100 μL of 30% solution of acetic acid (Chempur, Poland) was added and the plate was incubated at RT with shaking for 30 min. The solution was transferred to a separate plate and absorbance was measured spectrometrically at the wavelength λ = 550 nm. The reduction of biofilm cells was calculated as it was presented for the MIC evaluation. There were two replications performed for each method.

#### 4.3.3. Bacterial Cellulose Carrier Preparation

The most active extracts were also examined by the disc diffusion method. Instead of standard paper discs, we applied bacterial cellulose polymers, obtained as follows (Krasowski et al. [53]).

A *Komagataeibacter xylinus* ATCC 53524 strain was cultured in stationary conditions for 7 days at 28 °C in a Herstin–Schramm (H–S) medium. The medium was composed of 2% glucose (*w*/*v*; Chempur, Poland), 0.5% yeast extract (*w*/*v*; VWR, USA), 0.5% bacto-pepton (*w*/*v*; Graso, Poland), 0.115% citric acid monohydricum (*w*/*v*; Stanlab, Poland), 0.27% Na_2_HPO_4_ (*w*/*v*; Chempur, Poland), 0.05% MgSO_4_* 7H_2_O (*w/v*; Chempur, Poland), and 1% ethanol (Chempur, Poland). Next, the cultures were shaken to remove the bacteria from the cellulose. To obtain 14 mm BC discs, the H–S medium was inoculated with the released bacteria in a 24-well culture plate (Nest, China) and incubated for 7 days at 28 °C. Subsequently, the BC discs were removed from the plate and purified with 0.1 M NaOH (Chempur, Poland) at 80 °C. The BC discs were then washed with double-distilled water until the neutral pH was obtained and sterilized in an autoclave. The average weight of a cellulose disc was approximately 0.75 g and 0.007 g for wet and dry (dried for 24 h at 37 °C) BC, respectively. To analyze the antimicrobial activity of the extracts, the BC discs were soaked with the extracts (0.75 mL, 0.9 mL, or 1 mL), 1 mL of saline (positive control), or the solvent and kept refrigerated for 24 h. The percentage concentration of the extracts absorbed in the BC discs was calculated by the formula:(2)$$\mathrm{Extract}\;\mathrm{concentration}\;[\%]=\frac{\mathrm{EV}}{\left(\mathrm{WBC}-\mathrm{DBC}\right)+\mathrm{EV}}\times 100\%$$Abbreviations: EV—volume of the extract [ml]; WBC—weight of the wet BC disc [g]; and DBC—weight of the dry BC disc [g].

#### 4.3.4. Analysis of the Antimicrobial Activity of Extracts Released from the Bacterial Cellulose Carrier

The 0.5 MF suspensions of the microorganisms were prepared and cultured onto Mueller–Hinton agar (Biomaxima, Lublin, Poland) plates using sterile cotton swabs. The soaked BC discs were placed onto the inoculated plates. The plates were incubated for 24 h at 37 °C and zones of growth inhibitions were measured in mm. Each extract was examined once for each strain.

### 4.4. Statistical Evaluation

To evaluate the impact of the plants’ extracts on bacteria growth, multiple linear regression (MLR) was applied. Compounds contained in plants’ extracts were considered as explanatory (independent) variables. Independent variables with *p* < 0.25 on the univariate analysis were entered into the model. Additionally, to find the most significant subset of variables, the forward selection was applied. In the case of high linear correlation (*R* > 0.8) on the univariate analysis between two independent variables, the independent variable with the higher R with the dependent variable was entered into the model. The redundancy of the variable was accessed using the variance inflation factor (VIF). The explanatory variables with VIF lower than 3 were included in the model. Models for which *p* < 0.05 were considered significant. To determine which predictor was the most important in the regression models, the standardized regression coefficient b and R2 incremental were considered [61]. For the estimation of statistically significant differences in compound contents (Table 1 and Table 2), Tukey’s test was applied. Means signed with the same letter were not significantly different (*p* > 0.05). All statistical analyses were performed using TIBCO Software Inc. (2017) Statistica (data analysis software system), version 13.

## 5. Conclusions

Comparative studies on antimicrobial properties of extracts from five different Papaveraceae species indicated *C. cheilanthifolia* to be the most effective against human pathogens such as *S. aureus*, *P. aeruginosa*, and *C. albicans*. No direct correlation can be found between the results of antimicrobial activity and the amount of compounds contained in the extracts. The roots of *C. majus*, followed by *C. cheilanthifolia* and *F. vaillantii*, had the most diverse phytochemical profile in terms of isoquinoline alkaloids, while their aerial parts were rich in polyphenolic compounds. However, the phytochemical composition does not seem to determine the strength of the antimicrobial activity of the tested extracts as relatively poorer *C. cava* and *C. pumila* extracts were found to be more effective in the fight with bacterial and fungal planktonic and biofilm cells.

## Figures and Tables

**Figure 1 molecules-26-04778-f001:**
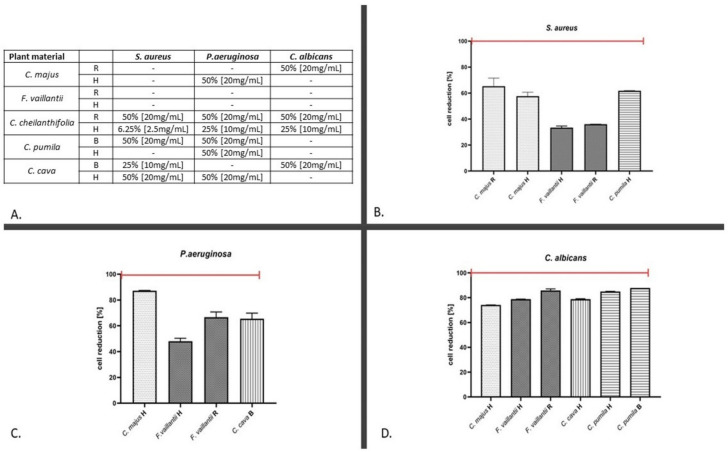
(**A**) Minimum inhibitory concentration (MIC (%)) of extracts against planktonic microbial cells. Dashes (−) indicate extracts in which the MIC values were not reached in the highest concentration (50%) of the extract applied. (**B**–**D**) Reduction of planktonic cell count below MIC’s threshold after the exposure to extracts, measured for *S.aureus, P.aeruginosa*, and *C.albicans* (in (**B**–**D**), respectively). Abbreviations: R—roots, H—herb, and B—bulbs.

**Figure 2 molecules-26-04778-f002:**
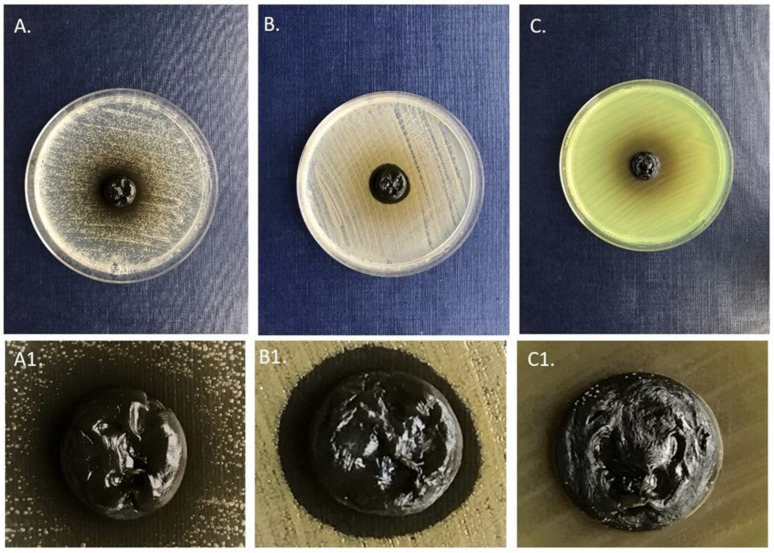
Zones of the inhibition of growth of (**A**) *C. albicans* and (**B**) *S. aureus* as the result of the release of the *C. cheilanthifolia* herb extract from the BC carrier. Please note that in the case of (**C**) *P. aeruginosa*, no inhibition zone was observed despite visible release (oval zone of dark color) of extract from the BC carrier. Pictures (**A1**–**C1**) are at 3x magnification of the BC carriers and inhibition zones (in the case of picture (**A1**,**B1**), and lack of zone in the case of (**C1**)) from pictures (**A**–**C**), respectively.

**Table 1 molecules-26-04778-t001:** Content [µg/g D.W. ± SD] of detected compounds in the aerial parts of *C. majus*, *C. cava*, *C. pumila*, *C. cheilanthifolia*, and *F. vaillantii*.

Number	Compound	Parent Ion (*m*/*z*)	Production (*m*/*z*)	Ion Mode	Content [µg/g D.W. ± SD]				
	ALKALOIDS				*C. majus*	*C. cava*	*C. pumila*	*C. cheilanthifolia*	*F. vaillantii*
1.	Protopine derivative	354	320, 260, 196	+	p	p	nd	p	p
2.	Allocryptopine	369.6	352, 187.9, 290	+	22.77 ± 1.75 ^a^	6.46 ± 0.70 ^b^	49.14 ± 1.41 ^c^	265.06 ± 3.73 ^d^	5.96 ± 0.50 ^b^
3.	Coptisine	320.1	291.9, 204, 262	+	5455.79 ± 22.97 ^a^	1820.82 ± 9.90 ^b^	2.75 ± 0.05 ^c^	5794.90 ± 22.72 ^d^	76.73 ± 1.04 ^e^
4.	Berberine	336.4	320, 292, 321.1	+	83.02 ± 1.77 ^a^	1.21 ± 0.07 ^b^	60.70 ± 0.88 ^c^	2696.04 ± 12.05 ^d^	1.82 ± 0.05 ^b^
5.	Chelidonine derivative	370	356, 339	+	p	nd	nd	p	Nd
6.	Chelidonine	353.8	275, 189, 247	+	130.77 ± 2.80 ^a^	1.85 ± 0.06 ^b^	0.24 ± 0.01 ^b^	nd^b^	8.03 ± 0.07 ^c^
7.	Chelerythrine	348.1	332, 304, 333	+	22.55 ± 0.54 ^a^	1.25 ± 0.09 ^b^	4.78 ± 0.1 ^c^	21.15 ± 0.25 ^d^	2.04 ± 0.08 ^b^
8.	Tetrahydroberberine	340	176, 149	+	p	p	p	p	p
9.	Tetrahydrocoptisine	324	176, 149	+	p	p	p	p	p
10.	Coptisine derivative	324	190	+	p	p	p	p	p
11.	Sanguinarine	332.1	274.1, 316.9, 246	+	33.24 ± 1.67 ^a^	2.53 ± 0.06 ^b^	6.23 ± 0.06 ^c^	57.22 ± 0.44 ^d^	22.71 ± 0.3 ^e^
12.	Protopine	320.2	303.2, 107, 123.8	+	315.67 ± 6.06 ^a^	136.52 ± 2.80 ^b^	890.77 ± 1.97 ^c^	748.43 ± 4.88 ^d^	1083.21 ± 10.83 ^e^
	MISCELLANEOUS								
13.	Malic acid	133.1	115, 71	−	p	p	p	p	p
14.	Trans-aconitic acid	172.9	85, 129	−	p	p	p	p	p
15.	Quinic acid	191	85, 93	−	p	p	p	p	p
16.	Salicylic acid	137.3	93, 65, 44.8	−	3.05 ± 0.10	nd	nd	LOQ	nd
17.	Trans-caffeic acid	179.2	135, 134, 89	−	130.37 ± 4.06 ^a^	52.89 ± 0.92 ^b^	27.17 ± 1.27 ^c^	16.05 ± 0.20 ^d^	50.98 ± 1.84 ^b^
18.	Chlorogenic acid	353	191, 85, 92.9	−	159.84 ± 1.04 ^a^	750.97 ± 2.68 ^b^	nd ^c^	558.65 ± 11.03 ^d^	82.37 ± 2.25 ^e^
19.	*p*-Coumaric acid	163.2	119.1, 93.1, 117	−	78.89 ± 2.58 ^a^	10.04 ± 0.53 ^b^	11.56 ± 0.66 ^b^	1.05 ± 0.08 ^c^	42.51 ± 1.65 ^d^
20.	Vanillin	151.2	136, 91.8, 108	−	15.54 ± 0.93	nd	nd	nd	nd
21.	Quercetin	301.1	151, 65, 121	−	1.75 ± 0.11 ^a^	1.23 ± 0.02 ^a^	26.20 ± 0.32 ^b^	206.01 ± 1.13 ^c^	262.36 ± 1.04 ^d^
22.	Kaempferol	285	169	−	nd	nd	nd	LOQ	nd

Abbreviations: p—present, identification was based on mass spectra with no reference substances; nd—not detected; LOQ—limit of quantification; Means with the same superscript letter (a, b, c, d, or e) within rows do not differ statistically at the significance level *p* ≤ 0.05 in Tukey’s test.

**Table 2 molecules-26-04778-t002:** Content [µg/g D.W. ± SD] of detected compounds in the underground parts of *C. majus*, *C. cava*, *C. pumila*, *C. cheilanthifolia*, and *F. vaillantii*.

Number	Compound	Parent Ion (*m*/*z*)	Production (*m*/*z*)	Ion Mode		Content [µg/g D.W. ± SD]			
	ALKALOIDS				*C. majus*	*C. cava*	*C. pumila*	*C. cheilanthifolia*	*F. vaillantii*
1	Protopine derivative	354	320, 260, 196	+	p	LOQ	p	p	p
2	Allocryptopine	369.6	352, 187.9, 290	+	147.96 ± 5.53 ^a^	36.52 ± 1.25 ^b^	699.72 ± 3.72 ^c^	330.01 ± 2.46 ^d^	9.26 ± 0.94 ^e^
3	Coptisine	320.1	291.9, 204, 262	+	2744.21 ± 14.02 ^a^	143.26 ± 1.66 ^b^	10.24 ± 0.25 ^c^	3605.56 ± 12.92 ^d^	134.24 ± 2.22 ^b^
4	Berberine	336.4	320, 292, 321.1	+	345.45 ± 5.76 ^a^	16.85 ± 0.3 ^b^	739.54 ± 3.41 ^c^	176.03 ± 3.32 ^d^	1.81 ± 0.03 ^e^
5	Chelidonine derivative	370	356, 339	+	p	LOQ	p	nd	nd
6	Chelidonine	353.8	275, 189, 247	+	1936.11 ± 15.48 ^a^	13.69 ± 0.85 ^b^	nd ^c^	6.74 ± 0.07 ^bc^	48.58 ± 1.28 ^d^
7	Chelerythrine	348.1	332, 304, 333	+	742.36 ± 4.63 ^a^	11.26 ± 0.24 ^b^	36.14 ± 0.89 ^c^	48.28 ± 1.10 ^d^	2.07 ± 0.06 ^e^
8	Tetrahydroberberine	340	176, 149	+	p	p	p	p	p
9	Tetrahydrocoptisine	324	176, 149	+	p	p	p	p	p
10	Coptisine derivative	324	190	+	p	p	p	p	p
11	Sanguinarine	332.1	274.1, 316.9, 246	+	915.56 ± 3.45 ^a^	17.46 ± 0.49 ^b^	28.30 ± 0.42 ^c^	493.20 ± 3.87 ^d^	45.56 ± 0.67 ^e^
12	Protopine	320.2	303.2, 107, 123.8	+	586.48 ± 12.67 ^a^	131.79 ± 1.56 ^b^	1260.62 ± 13.93 ^c^	6529.64 ± 13.88 ^c^	3610.09 ± 9.71 ^d^
	MISCELLANEOUS								
13.	Malic acid	133.1	115, 71	−	p	p	p	p	p
14.	Trans-aconitic acid	172.9	85, 129	−	p	p	p	p	p
15.	Quinic acid	191	85, 93	−	p	p	p	p	p
16.	Salicylic acid	137.3	93, 65, 44.8	−	nd	nd	nd	nd	LOQ
17.	Trans-caffeic acid	179.2	135, 134, 89	−	nd	LOQ	nd	nd	nd
18.	Chlorogenic acid	353	191, 85, 92.9	−	nd ^a^	1198.03 ± 8.63 ^b^	nd ^a^	51.18 ± 2.75 ^c^	1.22 ± 0.06 ^a^
19.	Vanillin	151.2	136, 91.8, 108	−	nd ^a^	nd ^a^	12.4 ± 0.28 ^b^	28.93 ± 1.03 ^c^	18.71 ± 0.94 ^d^
20.	Quercetin	301.1	151, 65, 121	−	nd ^a^	nd ^a^	nd ^a^	205.54 ± 1.19 ^b^	305.65 ± 1.78 ^c^

Abbreviations: p—present, identification was based on mass spectra with no reference substances; nd—not detected; LOQ—limit of quantification; Means with the same letter (a, b, c, d, e) within rows do not differ statistically at the significance level *p* ≤ 0.05 in Tukey’s test.

**Table 3 molecules-26-04778-t003:** Percentage reduction (%) of biofilm cells treated with extracts in the concentration of 50% (*v*/*v*) using the TTC and crystal violet (CV) method. The untreated control setting was established as 100% of microbial bacterial activity or biofilm biomass (method using TTC and CV, respectively). Please note that in the case of *P.aeruginosa*, the exposure on extracts led to an increase of metabolic activity measured by the TTC method; in the case of the biofilm assessment using the crystal violet assay, the results are not shown (n.s.) due to the low usability of this particular plate-based technique for pseudomonal biofilm analysis, as discussed in the later part of this manuscript. Abbreviations: R—roots, H—herb, and B—bulbs.

Plant Material		Reduction of Biofilm-forming Cell Number [TTC Assessment]			Reduction of Biofilm Biomass [Crystal Violet Assessment]		
		*S. aureus*	*P. aeruginosa*	*C. albicans*	*S. aureus*	*P. aeruginosa*	*C. albicans*
*C. majus*	R	71.79	−179.81	95.69	10.84	n.s	66.63
	H	26.11	−200.74	93.72	20.41	n.s	65.02
*F. vaillantii*	R	67.01	−97.61	92.92	24.29	n.s	66.20
	H	52.42	−94.27	95.04	20.81	n.s	65.27
*C. cheilanthifolia*	R	70.02	−177.17	93.55	7.63	n.s	74.64
	H	89.10	−194.71	93.85	10.99	n.s	68.92
*C. pumila*	B	66.84	−395.46	92.06	11.21	n.s	54.99
	H	82.74	−276.47	94.40	25.20	n.s	58.86
*C. cava*	B	67.61	−105.26	95.78	11.04	n.s	62.54
	H	74.53	−84.47	95.70	22.12	n.s	67.14

**Table 4 molecules-26-04778-t004:** Multiple regression-estimation of the influence of the variable on the model.

		*S. aureus*				*P. aeruginosa*				*C. albicans*			
		Compound	R^2^	Beta	*p*	Compound	R^2^	Beta	*p*	Compound	R^2^	Beta	*p*
**Aerial parts**	Planktonic cells/MIC	Quercetin	0.699	−0.614	0.000	Quercetin	0.965	−0.984	0.000	Berberine	0.960	−0.748	0.000
		Chlorogenic Acid	0.219	0.524	0.000					Coptisine	0.021	−0.268	0.003
		Sanguinarine	0.072	−0.322	0.001					Quercetin	0.010	−0.131	0.036
	Biofilm/TTC	*p*-coumaric Acid	0.781	−0.525	0.030	Exclusion of Data from Model				Chelerythrine	0.628	−0.818	0.004
	Biofilm/Crystal Violet	Statistical Model Insignificant				Statistical Model Insignificant				No Correlation *p* ˂ 0.25			
**Underground parts**	Planktonic Cells/MIC	Allocryptopine	0.340	0.892	0.001	Chelidonine	0.537	−0.487	0.020	Coptisine	0.825	−0.817	0.002
		Chlorogenic Acid	0.478	0.757	0.004	Allocryptopine	0.318	0.537	0.007				
	Biofilm/TTC	Sanguinarine	0.533	0.690	0.015	Allocryptopine	0.869	−0.869	0.002	Vanillin	0.478	−0.308	0.003
						Quercetin	0.033	0.192	0.000	Berberine	0.233	−0.233	0.005
										Sanguinarine	0.160	0.667	0.000
										Chlorogenic Acid	0.121	0.615	0.000
	Biofilm/Crystal Violet	Statistical Model Insignificant				Exclusion of Data from Model				Statistical Model Insignificant			

**Table 5 molecules-26-04778-t005:** Diameters of growth inhibition obtained by the disc diffusion method (mm). The diameters of the cellulose membranes were 14 mm. The release of extracts other than those obtained from *C. cheilanthifolia* H and *C. cava* H did not translate into the formation of the zone of growth inhibition.

Plant Material	Volume of Extract [mL]	Concentration of Extract [mg/mL]	Zone of Growth Inhibition [mm]		
			*S. aureus*	*P. aeruginosa*	*C. albicans*
*C. cava* H	0.9	21.76	19	0	0
*C. cheilanthifolia* H	1	22.8	21	22	0

## Data Availability

Data on the phytochemical analyses, antimicrobial assays as well as statistical evaluation are available from the authors.

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
