# Peer review of "Screening Papaveraceae as Novel Antibiofilm Natural-Based Agents"

_molecules, 2021, doi:10.3390/molecules26164778_

Round 1

Reviewer 1 Report

The present study described the phytochemical composition and antimicrobial properties of extracts from five different Papaveraceae species. The extracts have been chemisorbed within polymeric carrier to show their potential applicability as antimicrobial measure against localized infections.

The manuscript is needed some minor changes, as follow:

  1. The font of the generic and species names in the Abstract, Keywords and Results should be italic:

“Chelidonium majus, Corydalis cava, C. cheilanthifolia, C. pumila, and Fumaria vaillantii” -  Chelidonium majus, Corydalis cava, C. cheilanthifolia, C. pumila and Fumaria vaillantii.

 “S. aureus, P. aeruginosa and C. albicans” - S. aureus, P. aeruginosa and C. albicans

  1. In the introduction, some words are interrupted by a hyphen:

line 47:  „querce-tin” should be quercetin

line 49: “conclu-sively” should be conclusively

line 50, 52, etc.

  1. Grammatical errors:

Line 431: “isoiuinoline” should be isoquinoline

  1. For Compounds (1, 5, 8, 9, 10, 13, 14, 15) whose identifications are based only on mass spectra without reference substance, it is necessary to add literature data on their mass spectral fragmentation.

Author Response

Responses to Reviewers

The authors thank all the Reviewers for all kind comments and the valuable suggestions that have helped to improve the manuscript.

Reviewer 1

Comments and Suggestions for Authors

The present study described the phytochemical composition and antimicrobial properties of extracts from five different Papaveraceae species. The extracts have been chemisorbed within polymeric carrier to show their potential applicability as antimicrobial measure against localized infections.

The manuscript is needed some minor changes, as follow:

Q1: The font of the generic and species names in the Abstract, Keywords and Results should be italic:

“Chelidonium majus, Corydalis cava, C. cheilanthifolia, C. pumila, and Fumaria vaillantii” -  Chelidonium majus, Corydalis cava, C. cheilanthifolia, C. pumila and Fumaria vaillantii.

 “S. aureus, P. aeruginosa and C. albicans” - S. aureus, P. aeruginosa and C. albicans

Authors' response:  All Latin names of plant genera and species as well as microbial strains have been changed into italic throughout the mauscript.

Q 2: In the introduction, some words are interrupted by a hyphen:

line 47:  „querce-tin” should be quercetin

line 49: “conclu-sively” should be conclusively

line 50, 52, etc.

Authors' response:  All typo mistakes have been removed.

Q 3: Grammatical errors:

Line 431: “isoiuinoline” should be isoquinoline

Authors' response: This typo mistake has been removed.

Q 4: For Compounds (1, 5, 8, 9, 10, 13, 14, 15) whose identifications are based only on mass spectra without reference substance, it is necessary to add literature data on their mass spectral fragmentation.

Authors' response:  The appropriate literature used for the compounds identification (without reference substances) was inserted in the Materials and Methods section, ‘Identification and quantification’ subsection as follow:

„ The compounds identification was accomplished with the ion intensity ratio and retention time compared with the corresponding standards as well as with the previously identified alkaloids and phenolic acids reported in the literature (for compounds no 1, 5, 8, 9, 10 shown in Tables 1 and 2 - Grosso et al., 2014 [34]; for compounds no 13 and 14 – Bylund et al., 2007 [59]; for compound no 15 – Erk et al., 2009 [60]).”

Reviewer 2 Report

General comment

The manuscript describes on the screening Papaveraceae as novel antibiofilm natural-based agent with the objective to explore the similarities and differences in the 91 phytochemical profile and antimicrobial/antibiofilm potential of isoquinoline alkaloid 92 containing plants. Overall, the methodologies are robust, however, some clarification are still needed.

Specific comments

Slight Scientific English errors are noticeable.

Eg.

Page 18 line 609. Cristal violet to crystal violet

Page 17 line 579 and 18 line 607. 24h to 24 h

Page 18 line 612.  2h to 2 h.

And many more.

Page 17 line 578.  What is the initial concentration of the extract used for MIC? From that you can quantify how much extract used per well. Normally, MIC and MBC are reported for the concentration.  In your report, it is presented as %.  Is this refers to v/v or w/v? 

Page 8 line 262.  It is also suggested to the author to improve Figure 1A by replacing % to concentration.

Page 9 Table 3.  Crystal violet (cv) assay is used to determine the biofilm biomass (total microorganism and extracellular polysaccharides).  In the result showed n.s for cv assay but highly reduced biofilm in TTC assessment.  This is contradicting.  Did you run TTC alone without microorganism as a control.  I believe the data you have could be the background.  Please justify?

Page 17 line 580.  Please justify, why do you need to adjust to 0.5 McFarland first before standardise to 105 colony forming unit/mL?  Why don’t you straight away standardise? How do you standardised? Also, cells/mL is more appropriate since you are not counting the CFU.  Others, do you standardised all microorganisms (C. albicans, S. aureus and P. aeruginosa) to 105 cells/mL.  Candida has at leat 10 times the size of S. aureus, thus 0.5 McFarland will give different cell count between the yeast and bacteria.  Next, why do you use this ration between microorganism?  What are you mimicking?  Please justify.

Page 17 line 585. TTC is based on formazan which detect mitochondrial dehydrogenases enzyme.  The test is suitable to assess the metabolic activity for C. albicans as it is a eukaryotic microorganism.  However, for prokaryotes, there is no mitochondria presence.  Many studies reported on the inaccuracy of the test involving formazan on prokaryotes (bacteria).  Eg. Grela, E., KozÅ‚owska, J., & Grabowiecka, A. (2018). Current methodology of MTT assay in bacteria–A review. Acta histochemica120(4), 303-311.  Since Molecules is an high impact journal, thus, justification of the suitability of TTC method for bacteria in this study is needed.

Page 18 line 606.  How much is the concentration Eg. mg/mL?

Author Response

Responses to Reviewers

The authors thank all the Reviewers for all kind comments and the valuable suggestions that have helped to improve the manuscript.

Reviewer 2

Authors' response:

We would like to thank the Reviewer for the encouraging opinion and all thoughtful suggestions that have helped us improve the this manuscript. All changes in the manuscript, being result of the Reviewer’s suggestions, are presently highlighted green.  

General comment

The manuscript describes on the screening Papaveraceae as novel antibiofilm natural-based agent with the objective to explore the similarities and differences in the phytochemical profile and antimicrobial/antibiofilm potential of isoquinoline alkaloid containing plants. Overall, the methodologies are robust, however, some clarification are still needed.

Specific comments

Q 1: Slight Scientific English errors are noticeable.

Eg.

Page 18 line 609. Cristal violet to crystal violet

Page 17 line 579 and 18 line 607. 24h to 24 h

Page 18 line 612.  2h to 2 h.

And many more.

Authors' response: All typo mistakes have been corrected. Also, the manuscript has been proofread for English correctness and some minor changes have been made, as visible in the track changes mode of MS Word.

Q 2: Page 17 line 578.  What is the initial concentration of the extract used for MIC? From that you can quantify how much extract used per well. Normally, MIC and MBC are reported for the concentration.  In your report, it is presented as %.  Is this refers to v/v or w/v?

Authors' response: 

The working solution of extracts (100%) was 40 mg/ml, this information has been included in the manuscript, thank you for notification.

It refers to v/v, similarly as it is  presented when antiseptics activity are analyzed, where 100% (v/v) is “working solution” and  values <100% (typically 50%, 25%, 12,5%, etc.) are dilutions in medium containing microorganisms. Following your suggestion, we provided data on specific concentrations of extracts in appropriate paragraphs of manuscript, including Fig.1A.

Q 3: Page 8 line 262.  It is also suggested to the author to improve Figure 1A by replacing % to concentration.

Authors' response:  Thank you for this comment, we introduced concentration values in Figure 1A.

Q 4: Page 9 Table 3.  Crystal violet (cv) assay is used to determine the biofilm biomass (total microorganism and extracellular polysaccharides).  In the result showed n.s for cv assay but highly reduced biofilm in TTC assessment.  This is contradicting.  Did you run TTC alone without microorganism as a control.  I believe the data you have could be the background.  Please justify?

Authors' response: 

Authors thank for this remark. Yes, crystal violet stains microorganism regardless it's alive or dead. Contrary to that, TTC conversion to formazan takes place in the presence of living, but not dead microorganisms. Please see the Scanning Electron Microscopy picture we made of destructed staphylococcal cell in the biofilm - see pdf file of the responses.

Such a cell, with severely compromised cell wall integrity is still embedded within biofilm matrix. The CV will dye both matrix components and cell, while TTC conversion to formazan will be, in such case, significantly reduced. Such a phenomenon may lead to the observed result (no significant change in CV assessment between treated and non-treated biofilm; significant drop of metabolic activity in treated vs. non-treated biofilm).

In turn, in our study we observed such a phenomenon in case of P. aeruginosa biofilm. We are fully aware of the fact that majority of 96-well plate methods  are not fully comprehensive with analysis of the slime-forming biofilm of P. aeruginosa, and we highlighted this fact in Discussion part:

 “…The subsequent procedures of rinsing and washing, necessary to perform Cristal Violet assay, often lead to random removal of pseudomonal biofilm and to biased results of high standard deviations. Therefore, although we performed such an test, we deliberately did not include it to the manuscript. In turn, we presented data from metabolic (TTC) assay for analyzes of pseudomonal biofilm, because this technique allows, to some extent, avoid aforementioned disadvantages…”

Answering the part of question concerning impact of medium on TTC conversion to formazan; we are aware of fact that specific media, especially these containing high histidine content may, to some extent, facilitate  aforementioned chemical reduction, as it was indicated among others by:

Benov L (2019) Effect of growth media on the MTT colorimetric assay in bacteria. PLoS ONE 14(8): e0219713. https://doi.org/10.1371/journal.pone.0219713

Grela E, Kozłowska J, Grabowiecka A. Current methodology of MTT assay in bacteria - A review. Acta Histochem. 2018 May;120(4):303-311. doi: 10.1016/j.acthis.2018.03.007. Epub 2018 Mar 30. PMID: 29606555.

Nevertheless, in our study, when using TSB medium (Biomaxima, Poland) without microorganisms in the test conditions described in the Methodology Section we have never observed independent (without microorganism’s involvement) reduction of TTC into formazan.

Our earlier, internal experiments showed that such phenomenon may indeed occur, but it happened with no explicit pattern (i.e. it occurred in media of specific manufacturer(s) but with different LOT/batch number), therefore we assume that it may be related with animal- or plant-origin components of microbiological medium which is source of its variability.

Q 5: Page 17 line 580.  Please justify, why do you need to adjust to 0.5 McFarland first before standardise to 105 colony forming unit/mL?  Why don’t you straight away standardise? How do you standardised? Also, cells/mL is more appropriate since you are not counting the CFU.  Others, do you standardised all microorganisms (C. albicans, S. aureus and P. aeruginosa) to 105 cells/mL.  Candida has at leat 10 times the size of S. aureus, thus 0.5 McFarland will give different cell count between the yeast and bacteria.  Next, why do you use this ration between microorganism?  What are you mimicking?  Please justify.

Authors' response: 

Thank you for this important question. The overnight cultures of microorganisms multiply with various ratio, therefore the initial adjustment to 0.5 McFarland’s scale  (which is c.a. 1x108 cfu in case of bacteria and c.a. 1.5 x106 cfu in case of yeast-like strain) allows to dilute these overnight, dense cultures in single step to the concentrations that can be easily adjust to the desired value of 105 cfu. We would like to gently notice that in the discussed study we applied the three reference ATCC bacterial and fungal strains we are working with for c.a. 13 years and we performed calibration curve (correlation between McFarland value and cfu number) numerous times, please refer to our previously published papers:

 Brożyna, M.; Å»ywicka, A.; FijaÅ‚kowski, K.; Gorczyca, D.; Oleksy-Wawrzyniak, M.; Dydak, K.; MigdaÅ‚, P.; Dudek, B.; Bartoszewicz, M.; Junka, A. The Novel Quantitative Assay for Measuring the Antibiofilm Activity of Volatile Compounds (AntiBioVol). Appl. Sci. 2020, 10, 7343. https://doi.org/10.3390/app10207343

Krasowski, G.; Wicher-Dudek, R.; Paleczny, J.; Bil-Lula, I.; Fijałkowski, K.; Sedghizadeh, P.P.; Szymczyk, P.; Dudek, B.; Bartoszewicz, M.; Junka, A. Potential of Novel Bacterial Cellulose Dressings Chemisorbed with Antiseptics for the Treatment of Oral Biofilm Infections. Appl. Sci. 2019, 9, 5321. https://doi.org/10.3390/app9245321

Dudek-Wicher, R.K.; Szczęśniak-Sięga, B.M.; Wiglusz, R.J.; Janczak, J.; Bartoszewicz, M.; Junka, A.F. Evaluation of 1,2-Benzothiazine 1,1-Dioxide Derivatives In Vitro Activity towards Clinical-Relevant Microorganisms and Fibroblasts. Molecules 2020, 25, 3503. https://doi.org/10.3390/molecules25153503

The number of microorganisms equal of 105 cfu per gram of tissue (in some reports also 104 cfu) is believed to be the key indicator of potential “bioburden” as the causative factor associated with for example delayed wound healing.

  1. Malone, in The Microbiology of Skin, Soft Tissue, Bone and Joint Infections, 2017; (https://www.sciencedirect.com/topics/immunology-and-microbiology/bacterial-colonization).

The appropriate sections of manuscript have been changed according to Reviewer’s suggestion.

Q 6: Page 17 line 585. TTC is based on formazan which detect mitochondrial dehydrogenases enzyme.  The test is suitable to assess the metabolic activity for C. albicans as it is a eukaryotic microorganism.  However, for prokaryotes, there is no mitochondria presence.  Many studies reported on the inaccuracy of the test involving formazan on prokaryotes (bacteria).  Eg. Grela, E., KozÅ‚owska, J., & Grabowiecka, A. (2018). Current methodology of MTT assay in bacteria–A review. Acta histochemica, 120(4), 303-311.  Since Molecules is an high impact journal, thus, justification of the suitability of TTC method for bacteria in this study is needed.

Authors' response: 

Thank you for this insightful notification; we agree that the exact mechanism of bacterial ability to reduce TTC into formazan requires further investigation; we also are familiar with aforementioned work: Current methodology of MTT assay in bacteria–A review. We are aware that TTC methodology is imperfect and this is a reason for which we applied other antimicrobial tests (plate culturing, crystal violet assay), which are also, to some extent, biased and imperfect, but still they are considered major techniques for analyses of impact of various substances against microorganisms. We believe that application of various tests based on different principles (being at the same times aware of their specific flaws and limitations) allows to  draw proper conclusions. Nevertheless, to meet your demands and justify the use of TTC method, we allow ourselves to add the following sentence and quote the aforementioned work of Grela et al. in the appropriate Discussion part:

“As it was shown by Grela et al. [56] tetrazolium-based test, if performed with precautions, allows not only to localize and quantify biofilm  structure, but also to determine the presence of live bacteria within.”

Q 7: Page 18 line 606.  How much is the concentration Eg. mg/mL?

Authors' response:  The data on concentration has been provided, thank you for notice.

Reviewer 3 Report

Manuscript is well-written and the results support the conclusions. Accept the manuscript in its current form. 

Papaverae herbs have long been used medicinally for their antimicrobial properties. The bioactive substances of these plants interact in complex ways; however, the mechanisms remain unclear. The LC-ESI-MS/MS with three quadruple analyzers was used to analyze Chelidonium majus, Corydalis cava, Chelidonium cheilanthifolia, Chelidonium pumila, and Fumaria vaillantii minerals. The quality and quantity of all chemicals examined varied considerably from species to genera. Two types of metabolites dominated the phytochemical profile of the plant. It had polyphenolic and isoquinoline alkaloids. 22 and 21 chemicals were found in the aerial and sub-surface sections, respectively. Also, polyphenolic compounds were identified, such as malic acid, transaconite, quinic, salicylic, trans-caffeic, p-Coumaric, chlorogenic, quercetin, champferol, and vanilla. There were higher polyphenol levels in the air parts of all plants. Antibacterial activity was tested for S. aureus, P. aeruginosa, and Candida albicans in planktonic and biofilm-producing cells. Extracts were used to evaluate biofilm biomass generation. An antibacterial activity has also been tested on a Bacterial Cellulose polymer carrier. All other tested human infections exceeded C. cheilanthifolia extracts. However, the composition of other bioactive substances with the results of antimicrobial activity was not directly correlated. Continuous research must determine the relationships between known and unknown chemicals in extracts and biological characteristics. 

These findings are novel and warrant publication at Molecules.

Author Response

Responses to Reviewers

The authors thank all the Reviewers for all kind comments and the valuable suggestions that have helped to improve the manuscript.

Reviewer 3

Comments and Suggestions for Authors

Manuscript is well-written and the results support the conclusions. Accept the manuscript in its current form.

Papaverae herbs have long been used medicinally for their antimicrobial properties. The bioactive substances of these plants interact in complex ways; however, the mechanisms remain unclear. The LC-ESI-MS/MS with three quadruple analyzers was used to analyze Chelidonium majus, Corydalis cava, Chelidonium cheilanthifolia, Chelidonium pumila, and Fumaria vaillantii minerals. The quality and quantity of all chemicals examined varied considerably from species to genera. Two types of metabolites dominated the phytochemical profile of the plant. It had polyphenolic and isoquinoline alkaloids. 22 and 21 chemicals were found in the aerial and sub-surface sections, respectively. Also, polyphenolic compounds were identified, such as malic acid, transaconite, quinic, salicylic, trans-caffeic, p-Coumaric, chlorogenic, quercetin, champferol, and vanilla. There were higher polyphenol levels in the air parts of all plants. Antibacterial activity was tested for S. aureus, P. aeruginosa, and Candida albicans in planktonic and biofilm-producing cells. Extracts were used to evaluate biofilm biomass generation. An antibacterial activity has also been tested on a Bacterial Cellulose polymer carrier. All other tested human infections exceeded C. cheilanthifolia extracts. However, the composition of other bioactive substances with the results of antimicrobial activity was not directly correlated. Continuous research must determine the relationships between known and unknown chemicals in extracts and biological characteristics.

These findings are novel and warrant publication at Molecules.

The authors thank for the words of appreciation. The manuscript was re-checked for typo- and stylistic mistakes and supplemented with information requested by other reviewers. We hope that the introduced corrections have significantly improved the manuscript and that it will gain the appreciation of the reviewers in its present form.

Round 2

Reviewer 2 Report

The authors have improvised the manuscript and addressed all comments thoroughly.  The manuscript is ready for publication.